# LEARNING CAUSAL ALIGNMENT FOR RELIABLE DISEASE DIAGNOSIS

**Mingzhou Liu**[1] **Ching-Wen Lee**[1] **Xinwei Sun**[*2] **Xueqing Yu**[1] **Yu Qiao**[*3] **Yizhou Wang**[4,1,5,6,7]

[1] School of Computer Science, Peking University
[2] School of Data Science, Fudan University
[3] School of Automation and Intelligent Sensing, Shanghai Jiao Tong University
[4] Center on Frontiers of Computing Studies, Peking University
[5] Institute for Artificial Intelligence, Peking University
[6] Nat'l Eng. Research Center of Visual Technology, Peking University
[7] State Key Lab. of General Artificial Intelligence, Peking University
[*] Corresponding authors
{liumingzhou, jingwenli, yuxueqing}@stu.pku.edu.cn
sunxinwei@fudan.edu.cn, qiaoyu@sjtu.edu.cn, yizhou.wang@pku.edu.cn

## ABSTRACT

Aligning the decision-making process of machine learning algorithms with that of experienced radiologists is crucial for reliable diagnosis. While existing methods have attempted to align their diagnosis behaviors to those of radiologists reflected in the training data, this alignment is primarily associational rather than causal, resulting in pseudo-correlations that may not transfer well. In this paper, we propose a causality-based alignment framework towards aligning the model's decision process with that of experts. Specifically, we first employ counterfactual generation to identify the causal chain of model decisions. To align this causal chain with that of experts, we propose a causal alignment loss that enforces the model to focus on causal factors underlying each decision step in the whole causal chain. To optimize this loss that involves the counterfactual generator as an implicit function of the model's parameters, we employ the implicit function theorem equipped with the conjugate gradient method for efficient estimation. We demonstrate the effectiveness of our method on two medical diagnosis applications, showcasing faithful alignment to radiologists. Code is publicly available at https://github.com/lmz123321/Causal_alignment.

## 1 INTRODUCTION

Alignment is essential for developing reliable medical diagnosis systems Zhuang & Hadfield-Menell (2020). For instance, in lung cancer diagnosis, using models that are misaligned with clinical protocols can result in reliance on contextual features or instrument markers (Fig. 1 (c)) for diagnosis, leading to misdiagnosis and loss of timely treatment.

Despite the importance, alignment in medical imaging systems is largely understudied. Existing studies that are mostly related to us primarily focused on visual alignment, including Zhang et al. (2018); Chen et al. (2019); Brady et al. (2023) that proposed to learn object-centric representations, and Hind et al. (2019); Rieger et al. (2020) that adopted multi-task learning schemes to predict labels and expert decision bases simultaneously. Particularly, recent works Ross et al. (2017); Gao et al. (2022); Zhang et al. (2023) have proposed to regularize the model's input gradient to be within expert-annotated areas. However, their alignment with expert behaviors is only associational, rather than causal, making their models still biased towards spurious correlated features. This limitation is further explained in Fig. 1 (a), where two decision chains with different causal structures can exhibit similar correlation patterns.

In this paper, we propose a causal alignment approach that focuses on the alignment in the underlying causal mechanism of the decision-making process. Specifically, we first identify causal factors behind each decision step using counterfactual generation. We then propose a causal alignment loss

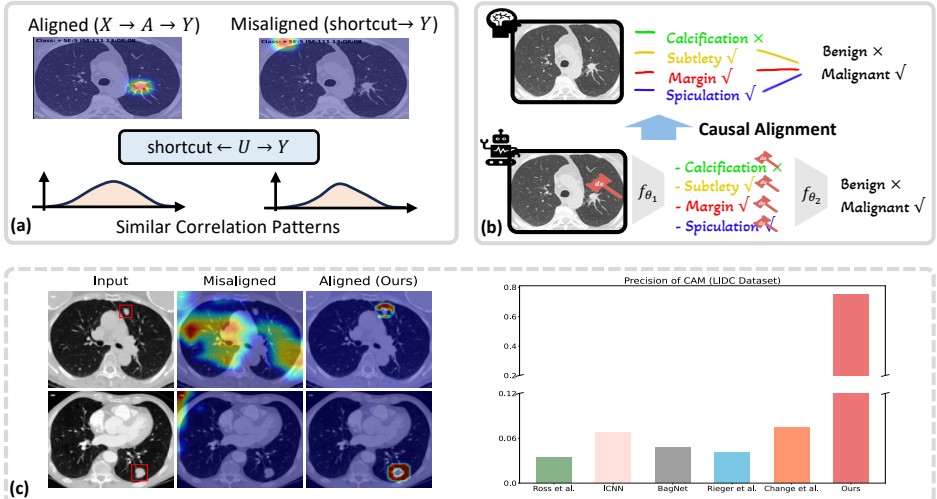

Figure 1: (a) Two decision chains with different causal structures but present similar correlation patterns. The left chain "mass $(X) \rightarrow$ attributes $(A) \rightarrow$ label $(Y)$" aligns with radiologists, while the right chain "shortcut $\rightarrow$ label $(Y)$" is misaligned. However, both $(X, Y)$ and (shortcut, $Y$) are correlated, due to the confounding bias between the shortcut and $Y$. (b) Our approach for learning causally aligned models. We first identify features and attributes that causally influence the model's decision, then align them to that of radiologists in a hierarchical fashion. (c) Class Activation Mapping (CAM) visualization and comparison of CAM precision on lung cancer diagnosis.

to enforce these identified causal factors to be aligned within those annotated by the radiologists. To optimize this loss that involves the counterfactual generator as an implicit function of the model parameters, we employ the implicit function theorem equipped with the conjugate gradient algorithm for efficient estimation. To illustrate, we consider the lung cancer diagnosis as shown in Fig. 1 (b). Guided by the alignment loss, our model can mimic the clinical decision pipeline, which first identifies the imaging area that describes attributes of the lesion, and then diagnoses based on these attributes Xie et al. (2020). Such training is facilitated by employing causal attribution Zhao et al. (2023) for inferring attributes that are causally related to the diagnosis. Returning to the lung cancer diagnosis example, Fig. 1 (c) shows that our method can learn causally aligned representations, in contrast to the features adopted by baseline methods, which are challenging to interpret.

**Contributions.** To summarize, our contributions are:

1. (**Causal alignment**) We propose a novel causal alignment approach to achieve alignment of causal mechanisms underlying the decision process of experienced radiologists.

2. (**Optimization**) We propose an efficient optimization algorithm by employing the implicit function theorem along with the conjugate gradient method.

3. (**Experiment**) We demonstrate the utility of our approach through significant improvements in alignment and diagnosis, on lung cancer and breast cancer diagnosis tasks.

## 2 RELATED WORKS

**Learning visual alignment.** Alignment is more broadly studied, *e.g.*, in natural language processing Ouyang et al. (2022) and reinforcement learning Ibarz et al. (2018). In the realm of visual alignment, Hind et al. (2019); Rieger et al. (2020) proposed to align deep learning models with humans by simultaneously predicting the class label and the decision area. Zhang et al. (2018); Liu et al. (2021b); Müller et al. (2023) aligned the decision-making process of neural networks by incorporating expert knowledge into architecture design. Of particular relevance to our work are Ross et al. (2017); Gao et al. (2022); Zhang et al. (2023), which suggested constraining the input image gradient to be significant in areas annotated by experts. However, the input gradient can be biased by pseudo-correlations that exist between expert features and shortcut features Geirhos et al. (2020), leading

to a misaligned model. In contrast, we adopt counterfactual generation to identify causal areas that determine the model's prediction. By ensuring these factors are confined to expert-annotated areas, our model can be effectively aligned with the expert's decision process.

**Explaining medical AI.** Explainability is essential for physicians to trust and utilize medical diagnosis models Lipton (2017). To achieve this, *attribution-based methods* explained model predictions by assessing the importance of different features Suryani et al. (2022); Yuen (2024). *Example-based methods* utilized similar images Barnett et al. (2021) or prototypes Gallée et al. (2024) to interpret the underlying decision rules. However, these approaches focused on interpreting models that have been trained. If misalignment occurs during the training process, their utility could be limited. In contrast, we propose an alignment loss to learn an intrinsically explainable model.

## 3 PROBLEM SETUP & BACKGROUND

In this section, we formulate our problem and introduce the background knowledge.

**Problem setup.** We consider the classification scenario, where the system contains an image $x \in \mathcal{X}$ and a label $y \in \mathcal{Y}$ from an expert annotator. In addition to $y$, we assume the expert also provides an explanation $e$ to explain his decision of labeling $x$ as $y$. Commonly, the explanation could refer to region of interest annotations or attribute descriptions. For example, radiologists often write an *annotation* section and an *observation* section, which respectively describe which body part is abnormal and what phenomena are observed, in their reports to explain their diagnosis Xie et al. (2020). Motivated by this, we assume for each sample, the explanation can be formulated as a binary mask $m$ indicating the abnormal area, along with a binary attribute description $a = [a_1, ..., a_p] \in \mathcal{A}$ of the abnormality. In this regard, our data can be denoted as $\mathcal{D} = \{(x_i, y_i, e_i = (m_i, a_i))\}_{i=1}^{n}$. With this data, **our goal** is then to learn a classifier $f_\theta : \mathcal{X} \mapsto \mathcal{Y}$ that **i)** predicts $y$ accurately **ii)** has a decision mechanism that is aligned with the radiologists.

**Structural counterfactuals.** To measure the likelihood that one event caused another, Pearl (2009) defines the following counterfactual quantity known as the *probability of causation*:

$$P(Y_x = y | X = x_0, Y = y_0), \tag{1}$$

which reads as "the probability of $Y$ would be $y$ had $X$ been $x$ if we factually observed that $X = x_0$ and $Y = y_0$"[1]. Here, $Y_x$ denotes the unit-counterfactual Pearl (2009) or potential outcome. In our scenario, rather than considering the whole image $x$, we are interested in specific regions within the image that causally determine the model's decision. To identify these regions, we adopt the following counterfactual generation scheme.

**Counterfactual (CF) Generation.** Given the classifier $f_\theta$ and any sample pair $(x_0, y_0)$, CF generates the counterfactual image $x^*$ with respect to the counterfactual class $y^* \neq y$ via Dhurandhar et al. (2018); Verma et al. (2020); Guyomard et al. (2023); Augustin et al. (2024):

$$x^* = \arg \min_x \mathcal{L}_{ce}(f_\theta(x), y^*) + \alpha d(x, x_0), \tag{2}$$

where $\mathcal{L}_{ce}$ is the cross-entropy loss for classification, $d(\cdot, \cdot)$ is a distance metric that constrains the modification to be sparse, and $\alpha$ is the regularization hyperparameter. In this regard, the modified area $supp(x^* - x_0)$ is responsible for the classification of $x_0$ as $y_0$, in that if we modified $x_0$ to $x^*$, the model would have made a different decision $y^*$. Indeed, in Prop. A.5, we can show that $x^*$ maximizes the probability of causation $P_\theta(Y_x = y^* | X = x_0, Y = y_0)$[2] induced by the classifier $f_\theta$, subject to $d(x, x_0) \leq d_\alpha$ for some $d_\alpha$.

To ensure the realism of the generated image, we can implement (2) using gradient descent in the image's latent space. Notably, this approach has proven effective for generating realistic images Goyal et al. (2019); Balasubramanian et al. (2020); Zemni et al. (2023), as also verified by the visualization of generated counterfactual images in Fig. 7.

Indeed, (2) is similar to but different from the optimization in **Adversarial Attack (AA)** Szegedy et al. (2013) concerning *perceptibility* Verma et al. (2020). Although both methods share the same objective framework, CF aims at highlighting significant areas that explain the classifier's decision

---

[1]Under the *exogeneity* and *monotonicity* conditions for binary $X, Y$, this quantity is identifiable.

[2]This term is identifiable since $f_\theta$ is known (see Prop. A.4 for details).

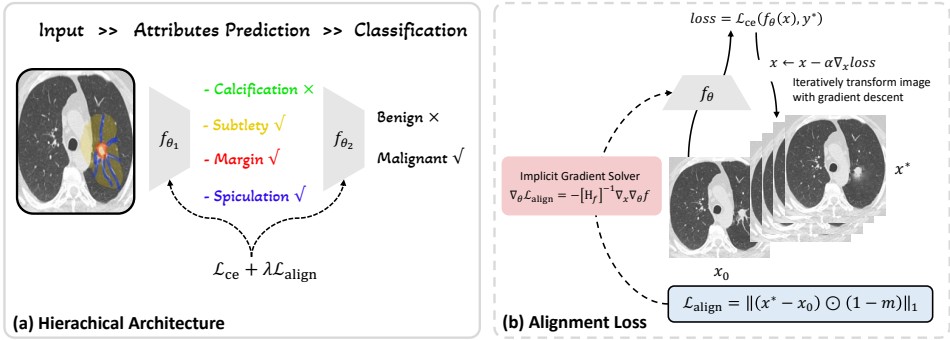

Figure 2: The schematic overview of our method. (a) We adopt a hierarchical structure that first provides attribute descriptions for the image, and then shows the diagnosis result. (b) Training with the proposed alignment loss. In the forward pass, a counterfactual image $x^*$ is generated and used to compute the alignment loss $\mathcal{L}_{\text{align}}$ relative to the expert's annotation $m$. In the backward pass, we use an implicit gradient solver to obtain the gradient $\nabla_\theta \mathcal{L}_{\text{align}}$ and use it to update the parameter $\theta$.

process, whereas AA favors making small and imperceptible changes to alter the prediction outcome Wachter et al. (2017). This often leads to different choices of the distance function $d(\cdot, \cdot)$ and the hyperparameter $\alpha$ Freiesleben (2022); Guidotti (2024).

## 4 METHODOLOGY

In this section, we introduce our framework for medical decision alignment. This section is composed of three parts. First, in Sec. 4.1, we introduce a *causal alignment loss* based on counterfactual generation, to align the model's decision bases with those of experts. Then, in Sec. 4.2, we propose to use the implicit function theorem equipped with conjugate gradient estimation to compute the gradient of our loss for optimization. Finally, in Sec. 4.3, we enhance our method with hierarchical alignment for cases where attribute annotations are available, using a hierarchical pipeline based on causal attribution. We summarize our framework in Fig. 2.

### 4.1 CASUAL ALIGNMENT LOSS

In this section, we propose a causal alignment loss to align the model with the experts. For illustration, we first consider the case where the attribute annotations are unavailable. The idea of our loss is to penalize the model once its counterfactual image contains modifications beyond radiologist-annotated areas. Specifically, we optimize a loss $\mathcal{L}_{\text{align}}$ of the following form:

$$\mathcal{L}_{\text{align}} := \frac{1}{n} \sum_{i=1}^{n} \left\| (x_i^* - x_i) \odot (1 - m_i) \right\|_{\ell_1}, \tag{3}$$

where $\odot$ denotes the element-wise matrix product, $x_i^*$ is the counterfactual image of $x_i$ obtained by (2), and $m \in \{0, 1\}^{\dim(x)}$ is the binary mask provided by radiologists. Then, by combining $\mathcal{L}_{\text{align}}$ with the cross-entropy loss for classification, we have our overall training objective:

$$\mathcal{L} = \mathcal{L}_{\text{ce}} + \lambda \mathcal{L}_{\text{align}},$$

where $\lambda$ is a tuning hyperparameter. To understand how the objective works towards alignment, note that $x^*$ maximizes the counterfactual likelihood $P_\theta(Y_x = y^* | x_0, y_0)$, indicating that $supp(x^* - x_0)$ represents the causal factors that influence the decision of the model $f_\theta$. Therefore, minimizing the distance between $supp(x^* - x_0)$ and $m$ encourages the model's causal factors to align more closely to those of the experts.

Our loss enjoys several advantages over alternative methods in visual alignment. Compared to Liu et al. (2021a;b); Müller et al. (2023) that incorporated prior knowledge into network architectures, our loss is more flexible and can be easily adapted to other scenarios and backbones. In contrast to Ross et al. (2017); Zhang et al. (2023) that constrained the input gradient, our approach can effectively avoid pseudo-features, benefiting from the identification of causal factors.

## 4.2 Optimization

In this section, we introduce the optimization process for the proposed alignment loss. For optimization, we need to compute the gradient $\nabla_\theta \mathcal{L}_{\text{align}}$, which involves the Jacobian matrix $\nabla_\theta x^*$. The main challenge here is that $x^*$ is an *implicit function* of $\theta$, defined by the argmin operator in (2), which makes it hard to compute $\nabla_\theta x^*$ explicitly.

To address this challenge, we resort to the Implicit Function Theorem (IFT), which allows us to compute the gradient in an implicit manner. Specifically, note that if $x^*$ is the minimum point of the function $T(x, \theta) := \mathcal{L}_{\text{ce}}(f_\theta(x), y^*) + \alpha d(x, x_0)$, it should satisfy that:

$$\nabla_x T \Big|_{x^*} = 0.$$

According to the law of total derivation, this implies that:

$$\nabla_\theta \nabla_x T \Big|_{x^*} = \left\{ \nabla_x(\nabla_x T) \cdot \nabla_\theta x^* + \nabla_\theta(\nabla_x T) \right\} \Big|_{x^*} = 0.$$

Therefore, computing $\nabla_\theta x^*$ boils down to the problem of solving the following linear equation:

$$H z^* = b, \tag{4}$$

where we denote $H := \nabla_x(\nabla_x T)$ as the Hessian matrix, $z^* := \nabla_\theta x^*$ as the Jacobian matrix, and $b := -\nabla_\theta(\nabla_x T)$ as the negative mixed derivative for brevity.

Formally speaking, we have the following theorem:

**Theorem 4.1** (Implicit Function Theorem (IFT), Krantz & Parks (2002))**.** *Consider two vectors* $x, \theta$*, and a differentiable function* $T(x, \theta)$*. Let* $x^* := \arg\min_x T(x, \theta)$*. Suppose that: **i)** the argmin is unique for each* $\theta$*, and **ii)** the Hessian matrix* $H$ *is invertible. Then* $x^*(\theta)$ *is a continuous function of* $\theta$*. Further, the Jacobian matrix* $\nabla_\theta x^*$ *satisfies the linear equation (4).*

Thm. 4.1 suggests that we can compute the Jacobian matrix using $\nabla_\theta x^* = -H^{-1}b$, which then gives $\nabla_\theta \mathcal{L}_{\text{align}}$ with the chain-rule. Nonetheless, for imaging tasks, typically $\theta$ is the parameter of high-dimensional neural networks, making it intractable to compute the Hessian matrix and its inverse. To address this issue, we employ the conjugate gradient algorithm Vishnoi et al. (2013) to estimate the solution of (4), without explicitly computing or storing the Hessian matrix. Notably, the conjugate gradient method has been successfully deployed in Hessian-free methods for deep learning Martens et al. (2010) and meta learning Sitzmann et al. (2020).

We briefly introduce the idea of conjugate gradient below, with a detailed discussion left to Vishnoi et al. (2013) (Chap. 6). To begin with, note that solving (4) is equivalent to solving:

$$z^* = \arg\min_z g(z), \text{ where } g(z) := \frac{1}{2} z^\top H z - b^\top z,$$

in that the minimum point $z^*$ satisfies $\nabla_z g \Big|_{z^*} = H z^* - b = 0$.

In this regard, we can implement gradient descent to minimize $g(\cdot)$, where the minimum point gives the solution of (4). During the minimization, the direction of the gradient updating is set to be conjugate (*i.e.*, orthogonal) to the residual $b - H z^{(i)}$, where $z^{(i)}$ is the estimate of $z^*$ in the $i$-th iteration, in order to achieve optimal convergence rate. To achieve this without explicitly forming $H$, we can leverage the *Hessian vector product* Song & Vicente (2022). Specifically, for $\epsilon$ that is a small perturbation around $z$, we have:

$$\nabla g(z + \epsilon z^{(i)}) \approx \nabla g(z) + H \epsilon z^{(i)}.$$

It then follows that:

$$H z^{(i)} \approx \frac{\nabla g(z + \epsilon z^{(i)}) - \nabla g(z)}{\epsilon},$$

which means we can estimate $H z^{(i)}$ with the finite difference of $\nabla g$ on the right-hand side.

Equipped with Thm. 4.1 especially the conjugate gradient method for estimation, we now summarize the optimization process for our loss in Alg. 1.

---

**Algorithm 1** Causal alignment training

---

**Input:** Data $\mathcal{D}$,
**Output:** Decision model $f_\theta$,
**Hyperparameters:** Sparsity regularization $\alpha$, weight of alignment loss $\lambda$, learning rate $\eta$.
 1: **while** not converged **do**
 2:     **\*\****Forward pass*
 3:         Compute $\mathcal{L}_{ce}$.
 4:         Optimize (2) to obtain $x^*$ and compute $\mathcal{L}_{align}$ using (3).
 5:         Compute $\mathcal{L} \leftarrow \mathcal{L}_{ce} + \lambda\mathcal{L}_{align}$.
 6:     **\*\****Back propagation*
 7:         Estimate $\nabla_\theta \mathcal{L}_{align}$ with conjugate gradient.
 8:         Update $\theta$: $\theta \leftarrow \theta - \eta\nabla_\theta\mathcal{L}$.   *// or Adam*
 9: **end while**

---

### 4.3 HIERARCHICAL ALIGNMENT

In this section, we extend our method to the scenario where attribute annotations are available. We introduce a hierarchical alignment framework to mimic the clinical diagnostic procedure.

**Causal diagram and assumptions.** We characterize this diagnostic process with the causal graph in Fig. 3. According to McNitt-Gray et al. (2007); Lee et al. (2017), the first step in the diagnosis is annotating each mass attribute from the image Xie et al. (2020). Therefore, we assume causal edges from the image $X$ to the attributes $A$. Since these attributes are directly annotated from the image, we assume no additional dependencies among them, implying their conditional independence given $X$. Building on these attributes, we further assume a causal relationship $A \rightarrow Y$, representing the decision-making process from the attributes to the final decision label.

Specifically, our classifier $f_\theta$ consists of an $f_{\theta_1} : \mathcal{X} \mapsto \mathcal{A}$ that predicts the attributes from the image $x$, and an $f_{\theta_2} : \mathcal{A} \mapsto \mathcal{Y}$ that classifies the label based on the predicted attributes. *For counterfactual generation*, we first find attributes responsible for predicting $y$ by altering the predicted attributes $\hat{a} := f_{\theta_1}(x)$ to the counterfactual ones $a^*$. Then, we locate image features that account for the modification of $|a^* - \hat{a}|$ via another counterfactual optimization over $x$ and obtain the counterfactual image $x^*$. For hierarchical alignment, we require both $|a^* - \hat{a}|$ and $|x^* - x|$ to be aligned with the expert's annotations of causal attributes and image regions, respectively.

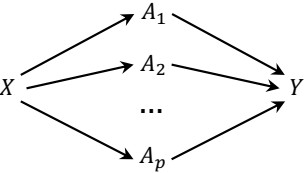

Figure 3: Causal diagram of radiologists' decision process. $A$ and $Y$ denote the expert's annotations of the attributes and the decision label, respectively.

**Causal attribution for annotations.** Although the attribute annotations can be available for many cases Armato III et al. (2011); Lee et al. (2017), it is hard to know which ones of these attributes causally determined the labeling of radiologists for each specific patient. To identify the causal attributes for alignment, we employ causal attribution based on counterfactual causal effect Zhao et al. (2023), which extends (1) to enable the quantification of the probability of causation for any subsets of attributes while conditioning on the entire attribute vector. Specifically, given evidence of the attributes $A = a$ and the label $Y = y$, we calculate the Conditional Counterfactual Causal Effect (CCCE) score for each attribute subset $S \subseteq \{1, ..., \dim(A)\}$:

$$\text{CCCE}(S) := \mathbb{E}(Y_{A_S=1} - Y_{A_S=0} | A = a, Y = y),$$

which is the difference between the conditional expectations of the potential outcomes $Y_{A_S=1}$ and $Y_{A_S=0}$ given the evidence. Recall that each attribute $A_i$ is binary. Then, according to Zhao et al. (2023) (Thm. 2), $\text{CCCE}(S)$ is identifiable and equals to

$$\text{CCCE}(S) \overset{(1)}{=} 1 - \frac{\text{P}\left(Y_{A_S=1} = y \mid A = a\right)}{\text{P}(Y = y \mid A = a)} \overset{(2)}{=} 1 - \frac{\text{P}\left(Y = y \mid A_S = 1, A_{-S} = a_{-S}\right)}{\text{P}(Y = y \mid A = a)},$$

where $A_{-S}$ denotes attributes beyond the subset $S$. Here, "(1)" arises from the exogeneity condition that there is no confounding between $A$ and $Y$ (*i.e.*, $Y_a \perp\!\!\!\perp A$), and "(2)" is based on the monotonicity

condition [3] that $Y_a \leq Y_{a'}$ if $a \preceq a'$ [4]. Both conditions are natural to hold in our scenario. Specifically, the exogeneity condition holds since the radiologist's decision $Y$ is based only on attributes (Fig. 3). For the monotonicity condition, it is easy to see that each intervention on any attribute from 0 to 1 (*e.g.*, from no speculation to speculation) will raise the probability of malignancy.

After computing the CCCE score for each attribute subset, we select the subset $S$ with the highest CCCE as the set of attributes causally related to the label. Accordingly, we set the annotation vector $r \in \{0,1\}^{\dim(A)}$ such that $r_S = (1, ..., 1)^\top$.

**Hierarchical alignment.** With such annotations, we introduce our hierarchical alignment process. Specifically, our objective function over $\theta = (\theta_1, \theta_2)$ is:

$$\mathcal{L}(\theta) := \mathcal{L}_{ce}(f_{\theta_2}(f_{\theta_1}(x)), y) + \mathcal{L}_{ce}(f_{\theta_1}(x), a) + \lambda_2 \mathcal{L}_{align}(\theta_2) + \lambda_1 \mathcal{L}_{align}(\theta_1), \quad (5)$$

where $\lambda_1 > 0, \lambda_2 > 0$ are tuning hyperparameters. Here, $\mathcal{L}_{ce}(f_{\theta_2}(f_{\theta_1}(x)), y)$ and $\mathcal{L}_{ce}(f_{\theta_1}(x), a)$ denote the cross-entropy losses for predicting $y$ and $a$, respectively.

The alignment loss $\mathcal{L}_{align}(\theta_2)$ over $\theta_2$ is defined as:

$$\mathcal{L}_{align}(\theta_2) := \frac{1}{n} \sum_{i=1}^{n} \|(a_i^*(\theta_2) - \hat{a}_i) \odot (1 - r_i)\|_{\ell_1},$$

where $\hat{a}_i := f_{\theta_1}(x_i)$ and the counterfactual attributes $a^*(\theta_2)$ is generated via:

$$a^*(\theta_2) = \arg\min_{a'} \mathcal{L}_{ce}(f_{\theta_2}(a'), y^*) + \alpha_2 d(a', \hat{a}). \quad (6)$$

Similarly, the alignment loss $\mathcal{L}_{align}(\theta_1)$ over $\theta_1$ is defined by (3), where the counterfactual image $x^*(\theta_1)$ that explains the change of $\hat{a}$ to $a^*$ is generated by:

$$x^*(\theta_1) = \arg\min_{x'} \mathcal{L}_{ce}(f_{\theta_1}(x'), a^*) + \alpha_1 d(x', x). \quad (7)$$

With the objective (5), we optimize $\theta$ by applying Alg. 1 to alignment terms. After the optimization, our decision process $x \rightarrow f_{\theta_2}(f_{\theta_1}(x))$ aligns well with that of the experts, with $f_{\theta_1}$ employing causal imaging factors to predict attributes, and $f_{\theta_2}$ using the causal attributes to predict $y$.

## 5 EXPERIMENT

In this section, we evaluate our method on two medical diagnosis tasks: the benign/malignant classification of lung nodules and breast masses.

### 5.1 EXPERIMENTAL SETUP

**Datasets & Preprocessing.** We consider the LIDC-IDRI dataset Armato III et al. (2011) for lung nodule classification and the CBIS-DDSM dataset Lee et al. (2017) for breast mass classification.

*The LIDC-IDRI dataset* contains thoracic CT images, each associated with bounding boxes indicating the nodule areas, six radiologist-annotated attributes (subtlety, calcification, margin, speculation, lobulation, and texture) and a malignancy score ranging from 1 to 5. Before analysis, we preprocess the images by resampling the pixel space and normalizing the intensity. We label those images with malignancy scores of 1-3 as benign ($y = 0$) and those with scores of 4-5 as malignant ($y = 1$). We split the dataset into training ($n = 731$), validation ($n = 238$), and test ($n = 244$) sets. *The CBIS-DDSM dataset* contains breast mammography images with fine-grained annotations (mass bounding boxes, attributes, and malignancy). We preprocess the images by removing the background and normalizing the intensity. We use the provided binary malignancy label and six annotated attributes (subtlety, shape, circumscription, obscuration, ill-definiteness, and spiculation). We follow the official dataset split, with 691 masses in the training set and 200 masses in the test set.

To test the ability of our method to learn expert-aligned features, we add a "+"/"−" symbol on the top-left corner of each image as a spuriously correlated feature. This symbol coincides with the

---

[3]Zhao et al. (2023) also assumed exogeneity condition and the monotonicity conditions among $A$, if there exist causal relations among $A$. Since there are no causal relations among $A$, we do not need these conditions.

[4]Here, $a$ and $a'$ are both vectors. $a \preceq a'$ denotes $a_k \leq a'_k$ for each $k$.

malignancy label in the training set, where images with $y = 1$ are labeled with "+" and those with $y = 0$ are labeled with "−"; but are assigned randomly in the validation and test sets. A well-aligned model should focus on the radiologist-annotated areas rather than the symbol.

**Evaluation metrics.** To assess the alignment of our model relative to radiologists, we compute the Class Activation Mapping (CAM) Selvaraju et al. (2017) and report its precision relative to the annotated areas, *i.e.*, $\frac{\text{Area of (CAM} \cap \text{Anno)}}{\text{Area of CAM}}$. We also report the overall classification accuracy.

**Implementation details.** We use the Adam optimizer and set the learning rate as $0.001$. We parameterize the attributes prediction network $f_{\theta_1}$ with a seven-layer Convolutional Neural Network (CNN), and train it for 100 epochs with a batch size of 128 for each iteration. For the classification network $f_{\theta_2}$, we parameterize it with a two-layer Multi-Layer Perceptron (MLP), and train it for 30 epochs with a batch size of 128. Please refer to Appx. B for details of the network architectures. For the hyperparameters $\alpha_1$ in (7) and $\alpha_2$ in (6), we set them to $\alpha_1 = 0.01, \alpha_2 = 0.0005$ for LIDC-IDRI and $\alpha_1 = 0.07, \alpha_2 = 0.0005$ for CBIS-DDSM, respectively. For both datasets, we set $\lambda_1 = \lambda_2 = 1$ in (5). For causal attribution, we calculate the CCCE scores of subsets containing no more than three attributes and select the subset with the highest score. We adopt the TorchOpt Ren et al. (2022) package to implement the conjugate gradient estimator. We repeat 3 different seeds to remove the effect of randomness.

## 5.2 COMPARISON WITH BASELINES

**Compared baselines.** We compare our method with the following baselines: **i) Ross et al. (2017)** that achieved interpretability by penalizing the input gradient to be small in object-irrelevant areas; **ii) ICNN** Zhang et al. (2018) that modified traditional CNN with an interpretable convolution layer to enforce object-centered representations; **iii) BagNet** Brendel & Bethge (2019) that approximated CNN with white-box bag-of-features models; **iv) Rieger et al. (2020)** that required the model to produce a classification as well as an explanation (*i.e.*, multi-tasks learning); **v) Chang et al. (2021)** that augmented the dataset with various factual and counterfactual images to alleviate the problem of learning spurious features; and **vi)** the **Oracle classifier** in which we manually restrict the input features to areas annotated by radiologists.

Table 1: Comparison with baseline methods on LIDC-IDRI and CBIS-DDSM datasets. The result of our method is **boldfaced** and the best result among baseline methods is underlined. For the Oracle classifier, the input features are manually restricted to the areas annotated by radiologists.

| Methodology | Precision of CAM | | Classification accuracy | |
|---|---|---|---|---|
| | LIDC | DDSM | LIDC | DDSM |
| Ross et al. (2017) | 0.034 (0.06) | 0.084 (0.11) | 0.656 (0.00) | 0.559 (0.05) |
| Zhang et al. (2018) | 0.068 (0.11) | 0.110 (0.13) | 0.381 (0.03) | 0.581 (0.00) |
| Brendel & Bethge (2019) | 0.048 (0.04) | 0.090 (0.04) | 0.358 (0.00) | 0.592 (0.00) |
| Rieger et al. (2020) | 0.041 (0.05) | 0.232 (0.17) | 0.343 (0.00) | 0.586 (0.01) |
| Chang et al. (2021) | 0.074 (0.03) | 0.119 (0.07) | 0.503 (0.08) | 0.496 (0.08) |
| Oracle classifier | 1.000 (0.00) | 1.000 (0.00) | 0.789 (0.00) | 0.726 (0.01) |
| Ours | **0.751 (0.03)** | **0.805 (0.06)** | **0.722 (0.00)** | **0.656 (0.00)** |

Due to the capability of capturing causal features, our method also significantly surpasses baseline models in terms of classification accuracy. This is due to the fact that, unlike the "+"/"−" symbol that demonstrates only spurious correlation to the label, features within the annotated areas have a causal relationship with the label, and therefore are transferable to test data.

Additionally, it is worth noting from Tab. 1 that even the oracle classifier only reaches a classification accuracy of 72% - 79%, which seems to contradict some previous results Wu et al. (2018); Wang et al. (2022); Liu et al. (2023) that claimed an accuracy of more than 99% in lung nodule classification and 90% in breast mass classification. To comprehend, this discrepancy is primarily due to the exclusion of challenging samples (those with a malignancy score of 3) in Wu et al. (2018), and the usage of custom training/test sets split in Wang et al. (2022); Liu et al. (2023).

## 5.3 ABLATION STUDY

In this section, we perform an ablation study on the causal alignment loss (Sec. 4.1) and the hierarchical alignment process (Sec.4.3). The results are shown in Tab. 2.

Table 2: Ablation study on LIDC-IDRI and CBIS-DDSM datasets.

| $\mathcal{L}_{\text{align}}$ | Hierarchical align | Precision of CAM | | Classification accuracy | |
|---|---|---|---|---|---|
| | | LIDC | DDSM | LIDC | DDSM |
| × | × | 0.057 (0.07) | 0.143 (0.20) | 0.535 (0.08) | 0.592 (0.00) |
| ✓ | × | 0.587 (0.08) | 0.621 (0.03) | 0.701 (0.02) | 0.633 (0.03) |
| ✓ | ✓ | **0.751 (0.03)** | **0.805 (0.06)** | **0.722 (0.00)** | **0.656 (0.00)** |

As we can see, both the alignment loss and the hierarchical procedure significantly improve the performance. In detail, the alignment loss accounts for a substantial portion of the improvement, yielding a 50% increase in CAM precision and a 15% boost in classification accuracy. Additionally, the hierarchical training strategy contributes an extra 20% to alignment precision and a 2% increase in classification performance. These results demonstrate the effectiveness of our alignment loss in learning features that coincide with radiologist assessments, as well as the significance of the hierarchical training strategy in mimicking the clinical diagnosis process.

## 5.4 VISUALIZATION

To further verify whether our method can learn radiologist-aligned features, we visualize the Class Activation Mapping (CAM) and show the results in Fig. 4.

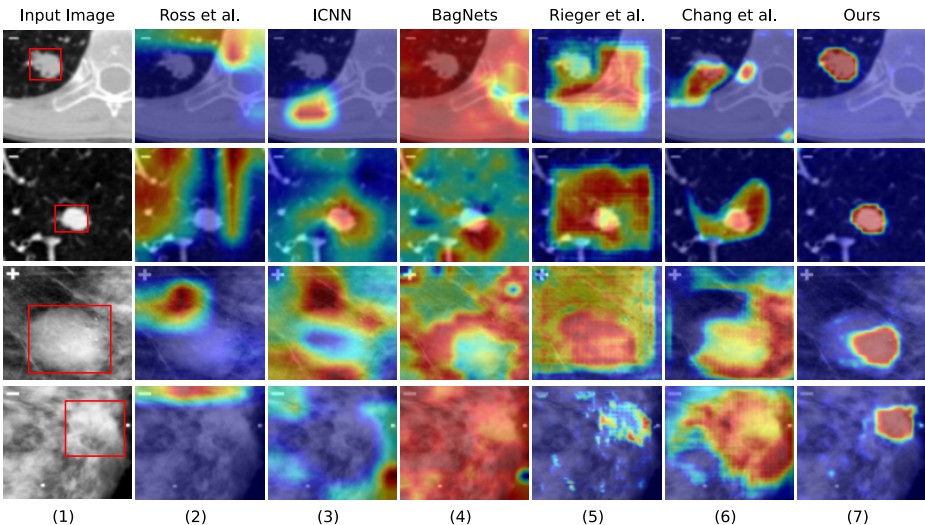

Figure 4: CAM visualization. Each row denotes different cases. The first column is the input images, where nodules and masses are marked by red bounding boxes. The second to seventh columns are CAMs of compared baselines and our method, respectively. See Appx. C.4 for more results.

As shown, the activation of our method concentrates on the nodule/mass areas, especially on the margins of the nodules/mass, which is a key feature for radiologists to evaluate the malignancy Sandler et al. (2023); Liu et al. (2024). In contrast, the activation of baseline methods focuses on lesion-irrelevant areas, such as the shortcut symbol "+"/"−" region in the top-left corner for Ross et al. (2017) and Brendel & Bethge (2019), or the background areas for Zhang et al. (2018), Rieger et al. (2020), and Chang et al. (2021). This visual analysis corroborates the quantitative results, demonstrating our method's ability to learn features that are well-aligned with the radiologist's diagnostic process.

## 6    CONCLUSION AND DISCUSSION

In this paper, we present a causal alignment framework to bridge the gap between the decision-making process of machine learning algorithms and experienced radiologists. By identifying the causal features that influence the model's decision, we can enforce the alignment of these causal areas with those of the radiologists through a causal alignment loss. This further allows us to train a hierarchical decision model that closely mirrors the expert's decision pipeline. The effectiveness of our approach is demonstrated by improved alignment in lung cancer and breast cancer diagnosis.

**Limitation and Future works.** The optimization of our causal alignment loss can be computationally expensive due to the estimation of the implicit Jacobian matrix. We will investigate efficient linear equation solving techniques Mou et al. (2016) to address this challenge. Additionally, we plan to apply our loss to alignment learning in multi-modality models and robotic systems.

## ACKNOWLEDGEMENTS

This work was supported by National Science and Technology Major Project (2022ZD0114904) and NSFC-6247070125; SJTU Trans-med Awards Research Fund 20220102; the Fundamental Research Funds for the Central Universities YG2023QNA47; NSFC-82470106. The authors thank Xuehai Pan for helpful discussions.

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

# Appendix

## A  CAUSAL ALIGNMENT THEORY

In this section, we discuss some theoretical aspects of causal alignment. We adopt the following notational convenience. Let $X$ and $Y$ denote the image and the predicted label, respectively. Let $Y_x$ denote the potential outcome of the predicted label under $X$ being $x$. Let $\phi_{\theta,\zeta}$ be the generation process that maps from the original image $x_0$ and the label $y_0$ to the counterfactual image $x$, which relies on the model parameter $\theta$ and random seed $\zeta$. We first require the following assumptions:

**Assumption A.1** (Consistency). We assume that for each individual, the predicted label $Y$ when $X = x$ is exactly the potential outcome $Y_x$.

**Assumption A.2.** We assume (2) has a unique global minimum solution.

*Remark* A.3. It can be shown that the global minimum of (2) can be attained via gradient descent under smoothness, and Polyak-Lojasiewicz conditions Polyak (1964); Csiba & Richtárik (2017). For deep learning optimization, the global minimum can be obtained if $f_\theta$ is over-parameterized Du et al. (2019) or has sufficient width Haeffele & Vidal (2017); Kawaguchi & Huang (2019).

Under the above assumptions, we show the probability of causation $P_\theta(Y_x = y | X = x_0, Y = y_0)$ is identifiable.

**Proposition A.4.** *Assume Asms. A.1, A.2, then the probability of causation is identifiable with*

$$P_\theta(Y_x = y | X = x_0, Y = y_0) = P_\theta(Y = y | x) P_\theta(x | x_0, y_0).$$

*Proof.* Denote the counterfactual generator as $\phi_{\theta,\zeta}$. If we fix the model parameter $\theta$ and the random seed $\zeta$, then $\phi_{\theta,\zeta}$ is a deterministic function, which means the conditional probability $P_\theta(x' | x_0, y_0) = \mathbb{1}(x' = \phi_{\theta,\zeta}(x_0, y_0)) = \mathbb{1}(x' = x)$ for any $x'$. Therefore, we have

$$P_\theta(Y_x = y | X = x_0, Y = y_0) = \int P_\theta(Y_x = y | x', x_0, y_0) P_\theta(x' | x_0, y_0) dx'$$
$$= P_\theta(Y_x = y | x, x_0, y_0) P_\theta(x | x_0, y_0).$$

Further, under the fixed seed $\zeta$, the potential outcome $Y_x$ is fully determined by the classifier $f_\theta$ and the counterfactual image $x$ via

$$Y_x = \text{sign}(f_\theta(x, u)),$$

where $u$ denotes the realization of the randomness $U$ in network prediction under the seed $\zeta$. Therefore, we have $Y_x \perp\!\!\!\perp (X_0, Y_0) | X = x$ and

$$P_\theta(Y_x = y | X = x_0, Y = y_0) = P_\theta(Y_x = y | x, x_0, y_0) P_\theta(x | x_0, y_0)$$
$$= P_\theta(Y_x = y | x) P_\theta(x | x_0, y_0)$$
$$= P_\theta(Y = y | x) P_\theta(x | x_0, y_0),$$

where the last equality is due to Asm. A.1. This shows the identification equation. $\square$

Below, we show $x^*$ in (2) maximizes the probability of causation, which means $supp(x^* - x_0)$ represents the causal factors that determine the model's decisions. As a result, minimizing $\mathcal{L}_{\text{align}}$ encourages the model's causal factors to align with those of the experts.

**Proposition A.5.** *Assume Asms. A.1, A.2, we have*

$$x^* = \arg\max_{x:d(x,x_0)\le d_\alpha} P_\theta(Y_x = y^* | X = x_0, Y = y_0)$$

*for some $d_\alpha$.*

*Proof.* **We first show** that (2) is equivalent to the following constrained optimization problem:

$$x^* = \arg\min_{x:d(x,x_0)\le d_\alpha} \mathcal{L}_{\text{ce}}(f_\theta(x), y^*). \tag{8}$$

To this end, let $d_\alpha := d(x^*, x_0)$ and let $x^\circ := \arg\min_{x:d(x,x_0)\le d_\alpha} \mathcal{L}_{\text{ce}}(f_\theta(x), y^*)$, we show

$$\mathcal{L}_{\text{ce}}(f_\theta(x^*), y^*) + \lambda d(x^*, x_0) = \mathcal{L}_{\text{ce}}(f_\theta(x^\circ), y^*) + \lambda d(x^\circ, x_0). \tag{9}$$

Since Asm. A.2 ensures the uniqueness of the minimum of (2), it then follows that $x^* = x^\circ$ and (8) holds. Now, note that $x^*$ satisfies $d(x^*, x_0) \leq d_\alpha$, which means

$$\mathcal{L}_{\text{ce}}(f_\theta(x^*), y^*) \geq \mathcal{L}_{\text{ce}}(f_\theta(x^\circ), y^*).$$

Since $x^\circ$ satisfies $d(x^\circ, x_0) \leq d_\alpha = d(x^*, x_0)$, we further have

$$\mathcal{L}_{\text{ce}}(f_\theta(x^*), y^*) + \lambda d(x^*, x_0) \geq \mathcal{L}_{\text{ce}}(f_\theta(x^\circ), y^*) + \lambda d(x^\circ, x_0).$$

Since $x^*$ minimizes (2), we also have

$$\mathcal{L}_{\text{ce}}(f_\theta(x^*), y^*) + \lambda d(x^*, x_0) \leq \mathcal{L}_{\text{ce}}(f_\theta(x^\circ), y^*) + \lambda d(x^\circ, x_0).$$

Therefore, we have (9) holds.

**We then show** $x^*$ maximize the probability of causation. From (8), we have

$$x^* = \arg\max_{x:d(x,x_0)\leq d_\alpha} \mathrm{P}_\theta(Y = y^*|x)\mathrm{P}_\theta(x|x_0, y_0),$$

where the term $\mathrm{P}_\theta(x|x_0, y_0) = \mathbb{1}(x = \phi_{\theta,\varsigma}(x_0, y_0))$ represents the generating process of $x$, and the term $\mathrm{P}_\theta(Y = y^*|x)$ represents maximizing the logarithm likelihood in the cross-entropy loss.

Then, according to the identification quantity of $\mathrm{P}_\theta(Y_x = y^*|X = x_0, Y = y_0)$ shown in Prop. A.4, we have

$$x^* = \arg\max_{x:d(x,x_0)\leq d_\alpha} \mathrm{P}_\theta(Y_x = y^*|X = x_0, Y = y_0).$$

This concludes the proof. □

## B  NETWORK ARCHITECTURES

In this section, we show the network architectures used in lung nodule classification (see Fig. 5) and breast mass classification (see Fig. 6).

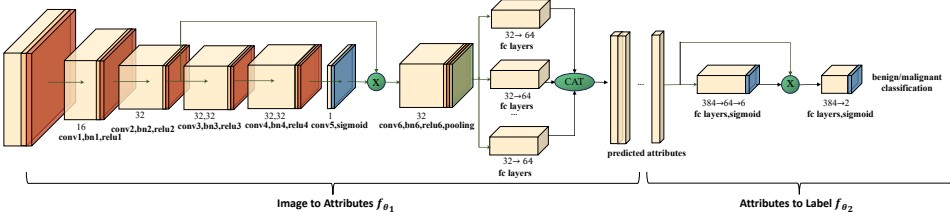

Figure 5: Network architecture used in lung nodule classification.

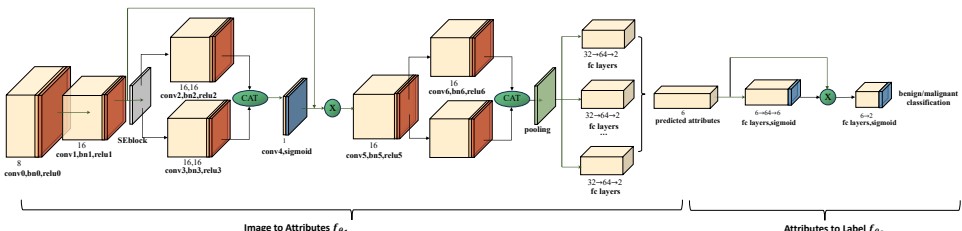

Figure 6: Network architecture used in breast mass classification.

## C  EXTRA EXPERIMENTAL RESULTS

### C.1  APPLICABILITY TO DIFFERENT DATA MODALITIES

In this section, we demonstrate the applicability of our method to different data modalities. Specifically, we consider brain MRI data from the BraTS dataset, breast ultrasound data from the Aryashah2k dataset, lung CT data from the LIDC-IDRI dataset Armato III et al. (2011), and breast mammogram data from the CBIS-DDSM dataset Lee et al. (2017). The results are presented in Tab. 3, showing that our method is consistently accurate across various data types.

Table 3: Performance of our method and baselines on different data modalities. The result of our method is **boldfaced** and the best result among baselines is underlined.

| Methodology | Precision of CAM | | | | Classification Accuracy | | | |
|---|---|---|---|---|---|---|---|---|
| | MRI | Ultra. | CT | Mamm. | MRI | Ultra. | CT | Mamm. |
| Ross et al. (2017) | 0.036 | 0.197 | 0.034 | 0.084 | 0.730 | 0.679 | 0.656 | 0.559 |
| Zhang et al. (2018) | 0.168 | 0.159 | 0.068 | 0.110 | 0.698 | 0.764 | 0.381 | 0.581 |
| Brendel & Bethge (2019) | 0.111 | 0.165 | 0.048 | 0.090 | 0.270 | 0.321 | 0.358 | 0.592 |
| Rieger et al. (2020) | 0.097 | 0.184 | 0.041 | 0.232 | 0.099 | 0.509 | 0.343 | 0.586 |
| Chang et al. (2021) | 0.147 | 0.127 | 0.074 | 0.119 | 0.410 | 0.270 | 0.503 | 0.496 |
| Ours | **0.908** | **0.872** | **0.751** | **0.805** | **0.835** | **0.797** | **0.722** | **0.656** |

### C.2  RESULTS UNDER DIFFERENT SHORTCUT SYMBOLS

We then show the performance of our method under various shortcut symbol settings. Specifically, we consider three cases: the +/- marker, intensity change, and the absence of a symbol. The results are presented in Tab. 4, showing that our method is effective across different shortcut symbols.

Table 4: Performance under different shortcut symbols.

| Symbol | Precision of CAM | | Classification Accuracy | |
|---|---|---|---|---|
| | LIDC | DDSM | LIDC | DDSM |
| None | 0.783 | 0.882 | 0.707 | 0.652 |
| Intensity | 0.760 | 0.783 | 0.723 | 0.670 |
| +/- | 0.751 | 0.805 | 0.722 | 0.656 |

## C.3 VISUALIZATION OF COUNTERFACTUAL IMAGES

In this section, we visualize the generated counterfactual images and show the result in Fig. 7. As we can see, the counterfactual modifications are clearly perceptible and align with specific clinical concepts, thereby validating the effectiveness of our counterfactual generation method.

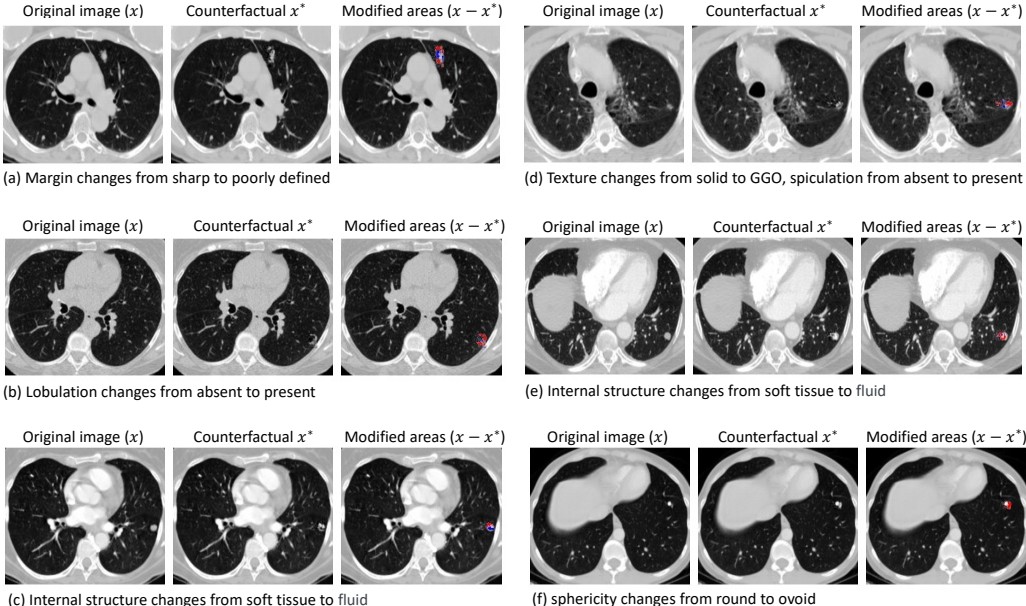

Figure 7: Generated counterfactual images on the LIDC-IDRI dataset. For each sub-figure, the left, middle, and right images denote the original image $x$, the counterfactual image $x^*$, and the modified area $supp(x - x^*)$, respectively. Positive modifications are marked in red and negative ones are marked in blue. We can observe that the counterfactual modifications all correspond to certain clinical attributes of the nodule, for example, in (a), the margin attribute changes from sharp to poorly defined when the label $y$ changes from benign to malignant.

## C.4 Visualization of CAMs

In this section, we provide more visualizations of the CAMs.

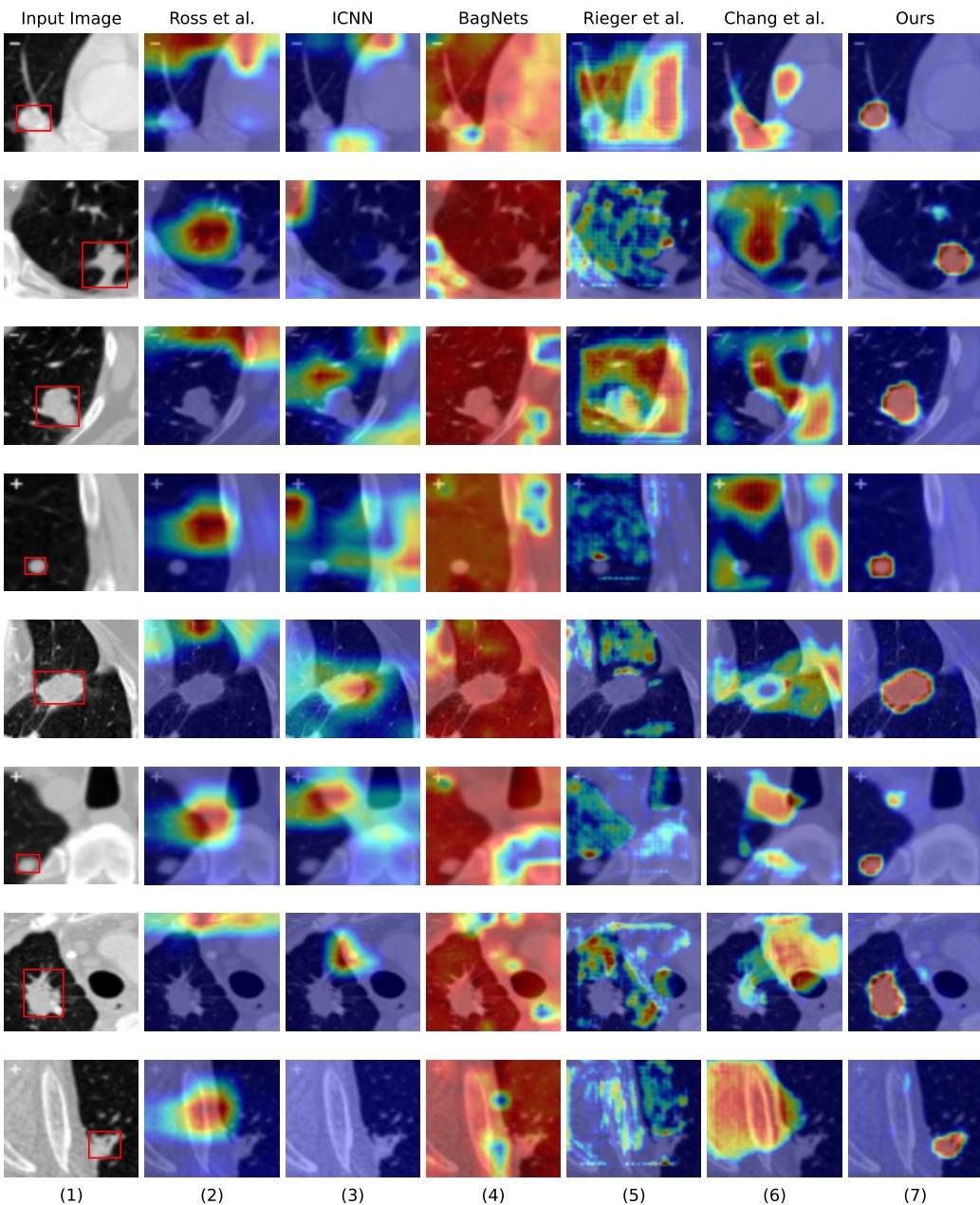

Figure 8: CAM visualization on the LIDC-IDRI dataset.

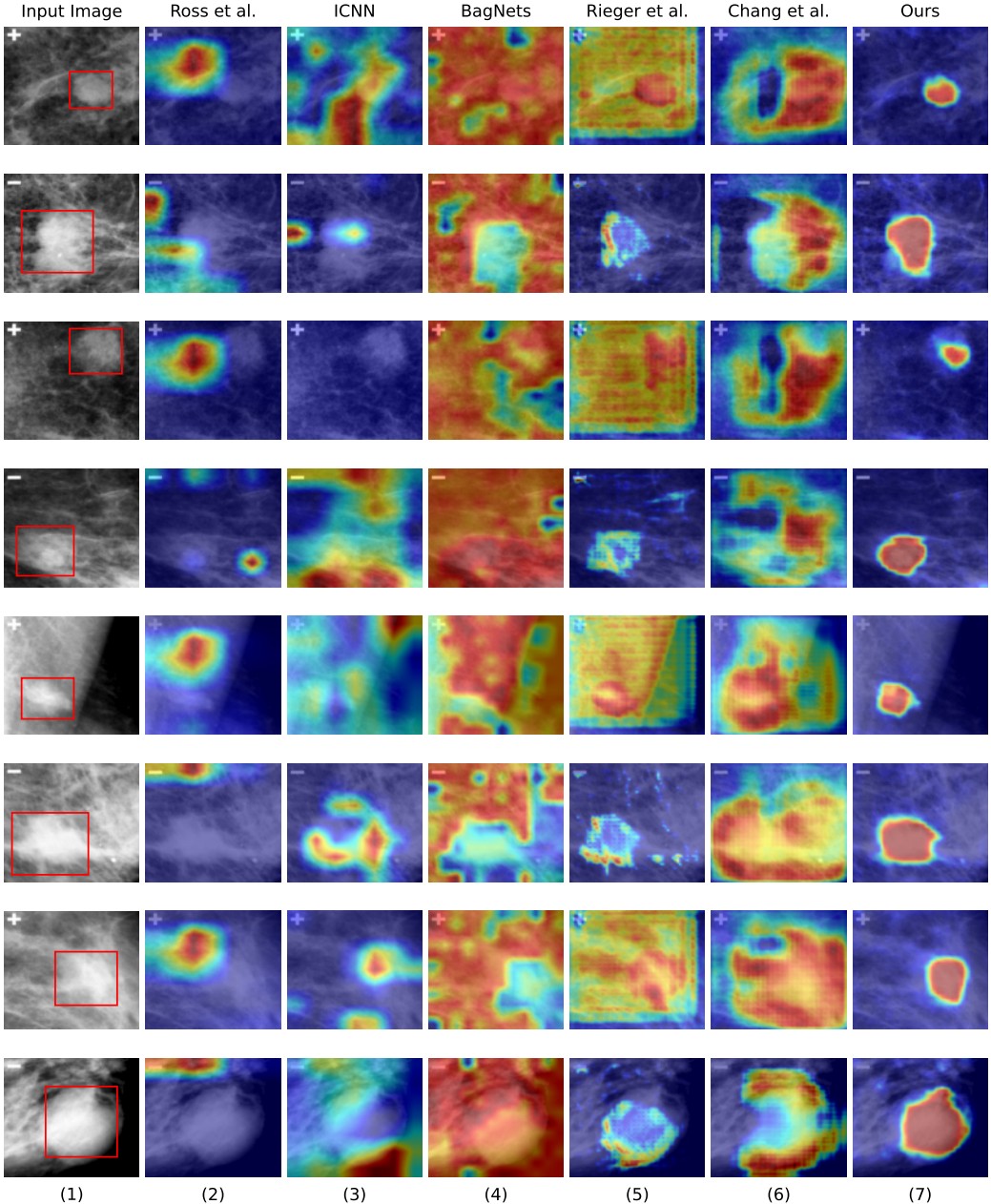

Figure 9: CAM visualization on the CBID-DDSM dataset.

