# OpenReview forum: "Learning Causal Alignment for Reliable Disease Diagnosis"
_ICLR.cc/2025/Conference — ICLR 2025 Poster_

### Official Review · Reviewer_DrbE · 2024-10-21

**Soundness:** 3
**Presentation:** 3
**Contribution:** 3
**Rating:** 6
**Confidence:** 3

**Summary:**

This work addresses the problem of identifying medical diagnoses while also uncovering factors that indeed causally contribute to these diagnoses. The authors develop a novel learning and optimization framework that efficiently and accurately achieves this goal. Empirical results demonstrate its superior performance compared to other methods.

**Strengths:**

High Novelty/Quality: This work develops a novel causal alignment loss and provides an efficient optimization method for solving the learning problem.

High Significance: The empirical results show that their approach not only achieves high diagnostic accuracy but, more importantly, identifies causal factors that contribute to the diagnosis. Accurate identification of causal factors is of paramount importance in the medical field, as it provides informed and precise information for planning subsequent treatments.

**Weaknesses:**

Clarity:

1. In Figure 2 and Figure 1.b, I am confused about the terms "calcification," "margin," and "spiculation." Are these attributes? From my understanding, which was described in the problem setup in section 3  the attributes a_i are binary indicators for the image. Do you mean that each of the binary attributes a_i, for example, indicates the presence of "calcification" in the image? It would be great if the authors could provide a more clear descriptions/ explanations in Figure 2 and Figure 1. b captions.

2. In Equation (2) for counterfactual generation, which distance function are you using? Since the choice of the distance function will indeed affect the downstream causal factor identification and diagnostic accuracy, it would be great if the authors could provide a section for discussion or ablation study to analyze the effect of distance function on the model performance and interpretability.

Besides common distance metrics (e.g., L1, L2, cosine similarity), possible prior-studied distance functions in the context of counterfactual generation are worthwhile for discussion [https://arxiv.org/pdf/1711.00399]:

Squared Euclidean Distance :
\begin{align} d(x_i,x^c_i) = \sum^d_{k=1} (x_{i,k} - x^c_{i,k})^2 \end{align}

Scaled Squared Euclidean Distance :
\begin{align} d(x_i,x^c_i) = \sum^d_{k=1} \frac{(x_{i,k} - x^c_{i,k})^2}{std_k(x)} \end{align}

Scaled L1 Norm :
\begin{align} d(x_i,x^c_i) = \sum^d_{k=1} \frac{|x_{i,k} - x^c_{i,k}|}{MAP_k} \quad \text{where,} \end{align} \begin{align} {MAP_k} = median_j (|x_{j,k} -median_l(x_{l,k})|) \end{align}

**Questions:**

Please see Weakness section.

**Details Of Ethics Concerns:**

This work is of important medical applications to help medical field in diagnosis and identifying causal factors of those diagnoses. It is very important for the authors to provide a paragraph in discussing their ethical implications.

---

> ### Author Response · Authors · 2024-11-18
> **Response to review DrbE**
>
> Thank you for acknowledging the novelty and significance of our article. We have modified the manuscript as suggested.
>
> **Regarding the attributes**. Yes, each of the binary variables $a_i$ indicates the presence or absence of an attribute.
>
> **Regarding the distance function**. We adopt the L1 norm in Eq. (2), which is also suggested by Wachter et al.
>
> In what follows, we conduct experiments to study the impact of different distance functions on counterfactual generation, with results presented in Fig. 7 in the updated manuscript. As we can see, the L1 norm encourages sparse modifications in critical features for malignancy assessment, such as spiculation and margin. In contrast, the squared Euclidean distance (L2 norm) tends to produce uniform changes across the whole nodule. This observation is consistent with the findings of Wachter et al. Moreover, we notice that the performance of L1/L2 norms and their scaled versions are similar, which can be attributed to the fact that we have already normalized the pixel values before training (see Sec. 5.1).
>
> We then show the performance of our method under different distance functions in the following table. The results indicate that the L1 norm outperforms the L2 norm, which can be attributed to the sparser modifications made by the L1 norm, facilitating more accurate localization of causal decision areas.
>
> | Distances  | Precision of CAM |  Classification accuracy |
> | --- | --- | --- |
> | L1 | 0.751 | 0.722  |
> | Scaled L1 | 0.714 |  0.702 |
> | L2 | 0.646  | 0.681 |
> |Scaled L2 | 0.640 | 0.658 |
>
> **Reference**:
>
> Wachter et al. Counterfactual explanation without opening the black box: automated decisions and the GDPR. https://arxiv.org/pdf/1711.00399

---

> > ### Comment · Reviewer_DrbE · 2024-11-19
> >
> > Dear Authors,
> >
> > Thank you so much for providing new experiments on the ablation study for the effect of the distance function on the metrics. It makes sense and I believe it is important to inform the readers that the L1 norm is the appropriate distance metric for interpretability, which leads to better accuracy.
> >
> > To follow up, by looking at Figure 7, it seems it is a bit difficult to observe L1 norm does not induce sparser modifications in critical features and the L2 induces uniform changes. Could you provide a quantifiable number for the L1/L2 comparison?
> >
> > Thanks.

---

> > > ### Author Response · Authors · 2024-11-21
> > > **Further response to review DrbE**
> > >
> > > Thank you for your response.
> > >
> > > We report the average number of pixels modified more than $10^{−3}$ under both the L1 and L2 norms in the table below, showing that the L1 norm indeed induces sparser modifications than the L2 norm.
> > >
> > > | L1 norm | L2 norm |
> > > | --- | --- |
> > > | 106.3 | 1146.5 |
> > > |  |  |

---

> > > > ### Comment · Reviewer_DrbE · 2024-11-21
> > > >
> > > > I appreciate it -- please include these numbers in your revision.
> > > >
> > > > Bests.
> > > >
> > > > Reviewer DrbE

---

> > > > > ### Author Response · Authors · 2024-11-22
> > > > > **Further response to review DrbE**
> > > > >
> > > > > We have incorporate this into the updated manuscript (Tab. 8). Thank you for your suggestion.

---

> > > > > > ### Comment · Reviewer_DrbE · 2024-11-23
> > > > > >
> > > > > > Dear Authors,
> > > > > >
> > > > > > As I mentioned earlier :
> > > > > >
> > > > > > "This work is of important medical applications to help medical field in diagnosis and identifying causal factors of those diagnoses. It is very important for the authors to provide a paragraph in discussing their ethical implications."
> > > > > >
> > > > > > Thanks.

---

> > > > > > > ### Author Response · Authors · 2024-11-24
> > > > > > > **Further response to review DrbE**
> > > > > > >
> > > > > > > We have incorporated the following ethical statement into the article. Thank you for your suggestion.
> > > > > > >
> > > > > > > **Ethical Statement**. The datasets used in this study were collected following ethical guidelines, including participant consent and anonymization, ensuring that there are no ethical concern or privacy issue. This paper proposes a causal alignment loss to align the decision-making processes of AI and experienced radiologists, significantly enhancing the credibility of AI-assisted diagnosis. Furthermore, the counterfactual generation and causal attribution methods employed in this study can identify the causal factors behind radiologist’s diagnosis, facilitating better understanding of the diagnosis procedure and the design of trustworthy medical AI.

---

> > > > > > > > ### Comment · Reviewer_DrbE · 2024-11-24
> > > > > > > >
> > > > > > > > Dear Authors,
> > > > > > > >
> > > > > > > > Thank you for your response. Although I believe your model faithfully identifies causal factors, I suggest mentioning that it is intended to be used as an assistance tool for radiologists. Additionally, you should either discuss the potential risks of using the tool without radiologists' supervision or emphasize once again that it is an AI-assisted tool.
> > > > > > > >
> > > > > > > > Thanks again.

---

> > > > > > > > > ### Author Response · Authors · 2024-11-24
> > > > > > > > >
> > > > > > > > > Thank you again for your suggestion. We have modified the ethical statement.
> > > > > > > > >
> > > > > > > > > **Ethical Statement**. The datasets for this study were collected under ethical guidelines, including participant consent and anonymization. This paper introduces a causal alignment loss to align AI decision-making with that of experienced radiologists, enhancing the credibility of AI-assisted diagnosis. Additionally, the counterfactual generation and causal attribution methods identify the causal factors behind radiologists‘ diagnoses, improving our understanding of the diagnostic process and facilitating the development of trustworthy medical AI. **Nonetheless, we emphasize that our method serves as an AI assistance tool for radiologists and should only be used under their supervision.**

---

> > > > > > > > > > ### Comment · Reviewer_DrbE · 2024-11-24
> > > > > > > > > >
> > > > > > > > > > Thanks authors for the prompt response and the modifications.
> > > > > > > > > >
> > > > > > > > > > Bests,
> > > > > > > > > >
> > > > > > > > > > Reviewer DrbE

---

### Official Review · Reviewer_bkax · 2024-10-29

**Soundness:** 4
**Presentation:** 4
**Contribution:** 4
**Rating:** 6
**Confidence:** 4

**Summary:**

The authors present a novel method to help researchers achieve alignment of causal attributions in image analysis for, e.g., lung disease, with that of experts. The novelty is that the alignment is grounded in causal attribution rather than mere association.

**Strengths:**

The results are compelling, and the problem it takes up is relevant and timely and promises immediate practical benefits if other researchers pursue it. Further elaboration of the idea shows promise.

The math used makes sense, given the causal model employed. I have some questions about the causal model that can be fixed in the presentation, which I describe below.

I am especially impressed with Figure 4, which shows that the method can infer regions of interest that are closely aligned with those of experts, and with Table 1, which shows the overall accuracy of the method.

In short, the argument that taking causal structure into account in the problem of aligning image analysis with experts in the biomedical sciences is well-put.

**Weaknesses:**

The main weakness in the paper, which, as I said, can I think be addressed textually, concerns the causal diagram in Figure 3. This diagram implies that the attributes selected by the procedure are independent of each other and conditional on the existence of a mass. There is no reason I can think of that this must be the case. For example, there could be a latent variable L such that X -> L -> {A1, A2}, in which case conditioning on X would not be sufficient to separate A1 and A2. Also, there's no reason to think that relevant features cannot overlap in the image, at least not in principle, in which case they will not be independent. Some nuance needs to be added to this point. I don't believe the model needs to be changed, but I feel it needs to be presented as a heuristic choice or a choice about the types of feature sets the authors aim to discover.

A second comment is that causal search algorithms are available that, with appropriate background knowledge, could also achieve a result of finding models in the form of Figure 3, and these have not been compared directly for efficacy. I will not recommend such models, as we are not supposed to in these reviews to recommend our own work. Still, the authors could search for these in the literature and explicitly state that perhaps, while they may be effective, a different approach is being taken up and evaluated here, or at least to give reasons why they are not pursued. Some of these methods are highly accurate in terms of the causal graphs that they infer, though the assumptions of these methods could be discussed briefly.

A third comment is that nothing rules out, in principle, the possibility that multiple masses might occur in a single image, and it is not clear what the method would say in cases like this.

A fourth comment is that what model class is being assumed here must be explicitly stated. It's a mixed linear Gaussian/discrete regime, though this should be stated, and if that's not the case, then that should especially be stated up front at the beginning of the article. If it's there and I missed it, then I apologize.

**Questions:**

Could you clarify the points listed in the Weaknesses section in the text and put the causal model here in the appropriate context?

---

> ### Author Response · Authors · 2024-11-18
> **Response to review bkax**
>
> We thank the reviewer for the highly constructive feedback and invaluable suggestions. We address your concerns below.
>
> **1**. "The main weakness in the paper, which, as I said, can be addressed textually, concerns the causal diagram in Figure 3. This diagram implies that the attributes are independent of each other conditional on the mass ... I don't believe the model needs to be changed, but I feel it needs to be presented as a heuristic choice or a choice about the types of feature sets the authors aim to discover."
>
> We are sorry for the confusion. Indeed, we choose the causal diagram in Fig. 3 to characterize the annotation process of radiologists. According to McNitt-Gray et al. (page 4) and Lee et al. (page 3), radiologists annotate each attribute solely based on image features, which explains the causal edges from $X$ to each $A_i$ and the conditional independence among attributes. We have incorporated this discussion to the article as suggested.
>
> **2**. "A second comment is that causal search algorithms are available for finding models in the form of Figure 3, and these have not been compared directly for efficacy.  The authors could search for these in the literature and explicitly state that perhaps, while they may be effective, a different approach is being taken up and evaluated here, or at least to give reasons why they are not pursued."
>
> We did not use causal discovery algorithms because our goal is to align the model's decision with that of radiologists. Therefore, we need to introduce the causal diagram that describes the decision process of radiologists (inspired by McNitt-Gray et al. and Lee et al.) ahead and use it as supervision for alignment.
>
> We also try the PC algorithm to recover the causal graph from data (see Fig. 8 in the updated manuscript) under the Markov and faithfulness assumptions. We find the skeleton of the recovered graph is consistent with that of Fig. 3.
>
> **3**. "Multiple masses might occur in a single image, and it is not clear what the method would say in cases like this."
>
> Our method can be adapted for cases with multiple masses. Specifically, if the goal is to classify the malignancy of the patient, we require the experts to additionally provide annotations indicating which masses causally contribute to the diagnosis. We can then apply our method for alignment. In the other case where the objective is to classify the malignancy of each mass, we can utilize existing mass detection algorithms (e.g., Armato et al.) to crop each mass from the images and apply our method to these cropped images.
>
>
> **4**. "A fourth comment is that what model class is being assumed here must be explicitly stated. It's a mixed linear Gaussian/discrete regime, though this should be stated, and if that's not the case, then that should especially be stated up front at the beginning of the article."
>
> Our method does not rely on specific parametric models, as both counterfactual generation and the CCCE are model-agnostic. They only require the assumptions outlined in Sec. 4.1 and Appx. A. We have incorporated this clarification into the article as recommended.
>
>
> **Reference**
>
> McNitt-Gray et al. The lung image database consortium data collection process for nodule detection and annotation.
>
> Lee et al. A curated mammography data set for use in computer-aided detection and diagnosis research.
>
> Armato et al. Lung cancer: performance of automated lung nodule detection applied to cancers missed in a CT screening program. Radiology.

---

> ### Author Response · Authors · 2024-11-21
> **Reminder of discussion**
>
> Thanks again for your efforts in reviewing our paper. We hope our responses address your questions and can thus help you reassess our paper. If you have additional questions, we'd be more than happy to provide additional clarification. Thank you for your attention.

---

> ### Author Response · Authors · 2024-11-24
>
> Dear reviewer
>
> As the deadline for discussion is approaching, would you mind reading our rebuttal and informing us whether we have addressed your concerns? If you have remaining concerns, we are more than happy for further discussion and clarification. Thanks for your attention.
>
> With regard, Authors

---

### Official Review · Reviewer_gNbm · 2024-11-01

**Soundness:** 3
**Presentation:** 4
**Contribution:** 3
**Rating:** 6
**Confidence:** 3

**Summary:**

This paper proposes a causality-based alignment framework to align the model’s decision with that of experts in disease diagnosis. The key components of the framework are counterfactual generation and causal alignment loss. Counterfactual generation highlights the areas that explain the model’s decision process, and the causal alignment loss minimized the distance between expert-annotated areas/attributes and the generated counterfactual. An optimization algorithm based on Implicit Function Theorem is designed to solve this two-stage problem. The authors then adapted the framework to two scenarios: area of interest annotations and attribute annotations, and develops a hierarchical alignment structure for attribute annotations. Various experiments are conducted and the results show that the causality-based alignment framework is robust against spurious correlation.

**Strengths:**

This is a good paper, I think. The presentation is concise, and the explanation of theory is clear. Numerical experiments are solid, with detailed description of implementation and comparison of baselines. Also, I think the idea of designing causal alignment loss based on counterfactual generation very fascinating.

**Weaknesses:**

My major question or concern for the causality-based alignment framework is computation cost, or scalability. For example, when solving for $x^*$, what if there are multiple counterfactual classes, or even more challenging, $y^*$ is continuous? I guess a straightforward solution would be to bin $y^*$ into binary classes, but that might sacrifice the granularity of counterfactual generation. Also, for hierarchical alignment, conditional counterfactual causal effect score is calculated for each attribute subset. What if the attributes are high-dimensional?

**Questions:**

Please see weakness.

---

> ### Author Response · Authors · 2024-11-18
> **Response to review gNbm**
>
> We thank the reviewer for the effort and positive feedback. Below, we address your concerns:
>
> **For multi-class scenarios**, we can simply chose a $y^*\neq y_0$, without needing to consider different values for $y^*$. This is because the alignment loss always constrains the classifier from altering regions beyond the expert annotations, no matter which $y^*$ is chosen.
>
>
> **For cases with a larger number of attributes**, we can first employ feature screening to select attributes that are dependent on $y$, and then compute the CCCE over the selected attributes. Nonetheless, we would like to point out that most medical diagnosis protocols, such as Lung-RADS and Bi-RADS, typically involve a limited number of attributes (usually fewer than ten).

---

> ### Author Response · Authors · 2024-11-21
> **Reminder of discussion**
>
> Thanks again for your efforts in reviewing our paper. We hope our responses address your questions and can thus help you reassess our paper. If you have additional questions, we'd be more than happy to provide additional clarification. Thank you for your attention.

---

> > ### Comment · Reviewer_gNbm · 2024-11-30
> >
> > Thank you for the authors' thoughtful response. I'm satisfied with the response.

---

### Official Review · Reviewer_SaCM · 2024-11-03

**Soundness:** 3
**Presentation:** 3
**Contribution:** 4
**Rating:** 6
**Confidence:** 3

**Summary:**

This paper addresses the problem of causal alignment, aiming to recover the causal mechanisms in a decision-making process. It specifically focuses on the classification of medical images, seeking to align the model's reasoning with the causal mechanisms used by experienced radiologists. With access to radiologist-annotated areas, the authors first use counterfactual generation to identify image regions that explain the classification result and then introduce a causal alignment loss to match these identified areas with the radiologist annotations. The alignment loss is optimized using the Implicit Function Theorem and the conjugate gradient algorithm. When attribute descriptions of abnormalities are available, they select the subset of attributes that causally determined the radiologists' labeling for each patient by maximizing the Conditional Counterfactual Causal Effect (CCCE). Finally, they propose a hierarchical alignment process to align both the annotated areas and attribute descriptions with expert annotations.

**Strengths:**

- To address the limitations in previous literature, where identified image regions align with expert behaviors only associationally, this paper proposes using counterfactual generation to identify regions that causally determine the model’s decision. The authors show that the generated counterfactual images maximize the probability of causation.
- To align the identified causal factors with radiologist-annotated areas, the paper introduces a causal alignment loss and an algorithm to estimate the gradient of this loss.
- The proposed method shows significant improvement over baseline methods in Class Activation Mapping (CAM) precision and classification accuracy. It effectively distinguishes spurious correlations from true causal factors.

**Weaknesses:**

- The selection of the set of attributes that are causally related to the label requires further justification. The paper claims that the subset $S$ maximizing the CCCE represents the causally related attributes. However, if an attribute $A_k$ is not a cause of $Y$ for some $k \in \{1:p\}$, then we would have $Y_{A_k = 1, A_{-k} = a_{-k}} = Y_{A_k = 0, A_{-k} = a_{-k}}$ for any fixed $a_{-k}$. Thus, the CCCE would be the same for the true causally related subset $S$ and any superset of $S$. It seems that the full set $A$ might also maximize CCCE. However, since only six attributes are annotated in both the LIDC-IDRI and CBIS-DDSM datasets, the proposed algorithm may still work even if $r_i = (1, \dots, 1)^T$ for all $i$.
- The paper claims that previous methods, which regularize the model's input gradient to align with expert-annotated areas (lines 37–40), only capture associational factors. However, the paper later (lines 405–407) attributes the failure of these methods not to their inability to distinguish spurious correlations, but to the limited effectiveness of gradient-based approaches themselves. In fact, even if the identified image regions are only associational factors, regularizing the gradient to expert annotations will rule out spurious correlations. Although the explanation in Figure 1(a) is conceptually reasonable, the simulation does not explicitly demonstrate the advantage of causal factors over associational factors when expert annotations are leveraged.
- It would be helpful to clarify Figure 3, the causal diagram that illustrates radiologists' decision process. It should be emphasized that $A_{1:p}$ represents the experts' "decision" about image attributes, rather than the underlying true attributes of the image. It initially appears to me that the underlying true attributes are inherent features of the image, in which case no causal relationship would exist between $X$ and these true attributes. Besides, how the area marks $m$ fit into the causal diagram could be explained further.
- Could the authors provide the computation time for the proposed method compared to the baseline methods?

**Questions:**

- Is the loss function in line 194 optimized directly? From Algorithm 1, it appears that this loss function is optimized in a two-step process, i.e. finding $x_i^*$ by minimizing $L_{ce}$ and then finding $\theta$ by minimizing $L_{align}$.

---

> ### Author Response · Authors · 2024-11-18
> **Response to review SaCM**
>
> We thank the reviewer for the positive assessment. We have modified the manuscript as suggested.
>
> **1**. "The selection of the set of attributes that are causally related to the label requires further justification ... However, since only six attributes are annotated in both the LIDC-IDRI and CBIS-DDSM datasets, the proposed algorithm may still work even if $r_i=(1,...,1)^\top$ for all $i$."
>
> We attribute only three causes to ensure that the selected features are informative.
>
> **2**. "The paper claims that previous methods, which regularize the model's input gradient to align with expert-annotated areas (lines 37–40), only capture associational factors. However, the paper later (lines 405–407) attributes the failure of these methods not to their inability to distinguish spurious correlations, but to the limited effectiveness of gradient-based approaches themselves."
>
> We apologize for any confusion. By “limited effectiveness of gradient-based approaches” mentioned in lines 405–407, we refer exactly to the fact that these methods capture only associational factors and do not offer causal explanations. This aligns with the findings in the paper cited in line 407 (Grimsley et al.). We have modified the manuscript for clarity.
>
> **3**. "In fact, even if the identified image regions are only associational factors, regularizing the gradient to expert annotations will rule out spurious correlations. Although the explanation in Figure 1(a) is conceptually reasonable, the simulation does not explicitly demonstrate the advantage of causal factors over associational factors when expert annotations are leveraged."
>
> We have compared with gradient-based methods (Ross et al.) in Tab. 1, showing significant advantages over associational features when expert annotations are leveraged. Could you please further elaborate on why the simulation does not illustrate the benefits of causal factors?
>
> **4**. "How the area marks $m$ fit into the causal diagram could be explained further."
>
> We have $X=$ {$X_m, X_{m^c}$}, where $X_m$ denotes the mask area and $X_{m^c}$ denotes the background area. The causal graph is then $X_m \to A \to Y; \,\, X_{m^c}$, where there is no causal edge connected to $X_{m^c}$.
>
> **5**. "Could the authors provide the computation time for the proposed method compared to the baseline methods?"
>
> The following table shows the computation costs of our method and baselines. As shown, and also noted in the limitations section, our method requires additional training time due to the estimation of the implicit gradient. However, considering the significant improvements in alignment and accuracy that are critical for medical diagnosis, this extra cost can be acceptable for practical applications.
>
> |    Methods  |   Time costs (h)   |
> | --- | --- |
> |   Ross et al.   |    1.5   |
> |   Zhang et al.  |  2.1   |
> | Brendel & Bethge | 1.3|
> | Change et al.| 1.6 |
> | Ours | 6.8 |
>
> **6**. "Is the loss function in line 194 optimized directly? From Algorithm 1, it appears that this loss function is optimized in a two-step process, i.e. finding $x^*$ by minimizing $\mathcal{L}_{ce}$ and then finding $\theta$ by minimizing $\mathcal{L}_{align}$."
>
> In Alg. 1, the last line should be “optimize $\theta$ with $\theta\leftarrow \eta \mathcal{L}$” instead of $\mathcal{L}_{align}$. We apologize for the typo. That means, the loss function in line 194 is optimized directly. Steps 2-4 calculate the gradient, and step 5 updates the parameter.

---

> > ### Comment · Reviewer_SaCM · 2024-11-20
> >
> > Thank you for providing the details on computational cost and the explanations of the methodology! These clarifications are really helpful. However, I still have some questions regarding my first point.
> >
> > I might be misunderstanding something, so I was hoping the authors could help me clarify this. I am still confused about why selecting the subset $S$ with the highest CCCE would necessarily yield the set of attributes causally related to the label. For example, if the true subset $S$ (the cause of the label) contains only 3 attributes, the full set $S'$ of 6 attributes should theoretically yield the same CCCE. Adding a non-causal attribute $A_k \in S' \backslash S$ would not affect the potential outcomes, so that $Y_{A_S = 1} - Y_{A_S = 0} = Y_{A_S' = 1} - Y_{A_S' = 0}$. I am curious whether the numerical results—where the subset $S$ maximizing CCCE contains only 3 attributes—are due to randomness in the estimation of CCCE. Could you elaborate on this?

---

> > > ### Author Response · Authors · 2024-11-21
> > > **Further response to review SaCM**
> > >
> > > Thank you for your response.
> > >
> > > Here, we limit our subsets to those containing $d = 3$ attributes, following Wang et al. and Miranda et al. This choice of $d$ is a balance between informativeness and comprehensiveness. Including too many attributes may lead to insufficient information for attribution, while selecting too few would fail to provide a sufficient explanation of the results. We have incorporate this discussion into the updated manuscript.
> > >
> > > References:
> > >
> > > Wang at al. Fine Grain Lung Nodule Diagnosis Based on CT Using 3D Convolutional Neural Network.
> > >
> > > Miranda et al. Computer-aided diagnosis system based on fuzzy logic for breast cancer categorization.

---

> > > > ### Comment · Reviewer_SaCM · 2024-11-21
> > > >
> > > > Thank you for your response. To clarify, my question was about why maximizing CCCE is able to separate the subset of attributes causally related to the label (as mentioned in lines 309-310), rather than why we need to select the subset.
> > > >
> > > > However, I don't think this point affects the main proposal regarding the alignment loss, which I find very interesting.

---

> > > > > ### Author Response · Authors · 2024-11-22
> > > > > **Further response to review SaCM**
> > > > >
> > > > > We are sorry for the confusion and would like to provide further clarification. CCCE focuses on causal attribution, whose goal is to identify underlying causes of observed events. Formally, it measures the counterfactual causal effect given an observed evidence, i.e., *would this event have happened, had this attribute been changed?* If the effect is large, it is more evident that this attributes causes that event.
> > > > >
> > > > > On the other hand, the cause-effect relation applies to the whole population under consideration. This relation must exist if it is observed in an occurred event; however, the presence of this relation does not mean it is the cause for some specific event. Specifically, if there is no edge from $A$ to $Y$, by noticing that $Y_{A \cup S} = Y_S$, we can conclude that adding $A$ to any subset $S$ would not increase the CCCE. However, the reverse may not be true. Even if there is a causal relation from $A$ to $Y$, it only implies an average effect of $A$ on $Y$, it may not be the cause for a specific event, which means adding $A$ does not necessarily increase the CCCE. For example, in lung cancer diagnosis, each attribute can have a significant effect on the disease; however, it can be that only a few attributes matter in a specific case. Since our goal is to identify attributes that causally determined the diagnosis, we calculate the counterfactual cause effect to attribute the decision for alignment.
> > > > >
> > > > > In the following table, we present the averaged maximum CCCE with respect to the number of attributes. Notably, the results indicate that having more attributes does not necessarily lead to a higher CCCE; for instance, the CCCE with six attributes is lower than that with three. Furthermore, the CCCE values for three, four, and five attributes are essentially similar, significantly higher than those with only one or two attributes. The discrepancy among them may be due to estimation errors. Therefore, in line with existing researches Wang et al. and Miranda et al., we restrict those subsets with only three attributes.
> > > > >
> > > > > | Number of attributes |  Averaged max CCCE |
> > > > > | --- | --- |
> > > > > |    1  |   0.553  |
> > > > > |    2  |   0.756  |
> > > > > |    3  |    0.816 |
> > > > > |    4 |   0.835  |
> > > > > |   5   |   0.837  |
> > > > > |   6   |    0.750 |

---

### Meta-Review · Area_Chair_LmXE · 2024-12-17

**Metareview:**

This is a "classical" borderline paper with both strengths and weaknesses. On the positive side, the idea of using counterfactual generation to identify regions that causally determine the model’s decision was considered interesting and novel.
On the negative side, the most severe concern seems to be the question about the effect of maximizing CCCE on separating the subset of attributes related to the labels. A related question concern the experimentally observed reduction of CCCE after removal of an attribute. Even after the rebuttal and discussion phase, these question could not be answered in a fully convincing way.   After going again over the rebuttal and all discussions, however, I think that this latter issue is probably not a deep conceptual problem, and that over-all, the positive aspects outweigh the negative ones.  Therefore, I recommend acceptance of this paper.

**Additional Comments On Reviewer Discussion:**

There was some discussion about the behavior about changes in CCCE and their potential role in separating the subset of attributes related to the labels. I have the impression, that the rebuttal could not fully address this issue in a fully convincing way. On the other hand, this open question does not seem to severely affect the over-all contribution of the paper and its novel ideas.

---

### Decision · Program_Chairs · 2025-01-22

Accept (Poster)